# Loneliness and its associated factors among university students during late stage of COVID-19 pandemic: An online cross-sectional study

**Wudneh Simegn** [1]*, **Gashaw Sisay**[2], **Abdulwase Mohammed Seid**[2], **Henok Dagne** [3]

**1** Department of Social and Administrative Pharmacy, School of Pharmacy, University of Gondar, Gondar, Ethiopia, **2** Department of Clinical Pharmacy, School of Pharmacy, University of Gondar, Gondar, Ethiopia, **3** Department of Environmental and Occupational Health and Safety, Institute of Public Health, University of Gondar, Gondar, Ethiopia

\* wudusim@gmail.com

## Abstract

### Background

The COVID-19 pandemic resulted in a number of psychosocial and emotional catastrophes, including loneliness. The associated lockdowns, reduced social support, and insufficiently perceived interactions are expected to heighten the level of loneliness during the pandemic. However, there is a dearth of evidence regarding the level of loneliness and what correlates with loneliness among university students in Africa, particularly in Ethiopia.

### Objectives

The general objective of this study was to assess the prevalence and associated factors of loneliness among university students during the COVID-19 pandemic in Ethiopia.

### Methods

A cross-sectional study was undertaken. An online data collection tool was distributed to voluntary undergraduate university students. The sampling technique used was snowball sampling. Students were requested to pass the online data collection tool to at least one of their friends to ease data collection. SPSS version 26.0 was used for data analysis. Both descriptive and inferential statistics were used to report the results. Binary logistic regression was used to identify factors associated with loneliness. A P-value less than 0.2 was used to screen variables for the multivariable analysis, and a P-value less than 0.05 was used to declare significance in the final multivariable logistic regression.

### Result

A total of 426 study participants responded. Out of the total, 62.9% were males, and 37.1% attended fields related to health. Over three-fourths (76.5%) of the study participants encountered loneliness. Females (adjusted odds ratio (AOR): 1.75; 95% confidence interval

**Data Availability Statement:** All relevant data are within the paper and its Supporting information files.

**Funding:** The author(s) received no specific funding for this work.

**Competing interests:** The authors have declared that no competing interests exist.

(CI): 1.01, 3.04), non-health-related departments (AOR: 1.94; 95% CI: 1.17, 3.35), ever encountering sexual harassment (AOR: 3.32; 95% CI: 1.46, 7.53), sleeping problems (AOR: 2.13; 95% CI: 1.06, 4.30), perceived stress (AOR: 6.40; 95% CI: 1.85, 22.19) and poor social support (AOR: 3.13; 95% CI: 1.10, 8.87) were significantly associated with loneliness.

## Conclusion and recommendation

A significant proportion of students were victims of loneliness during the COVID-19 pandemic. Being female, working in non-health-related fields, having sleeping problems, encountering sexual harassment, perceived stress, and poor social support were significantly associated with loneliness. Interventions to reduce loneliness should focus on related psychosocial support to reduce stress, sleeping disturbances, and poor social support. A special focus should also be given to female students.

## Introduction

Loneliness can be defined as a "distressing feeling that conveys the perception that one's social needs are not being met by the quantity and quality of one's social relationships" [1–3]. It is often described as a painful emotional experience of being without any desired social contact or in isolation from society [4–6]. Loneliness is considered a major public health problem among university students [7–9]. Loneliness due to the COVID-19 crisis results in higher problems when dealing with the habit [10]. It could result in worse physical and mental health problems [2, 11, 12] and increase mortality risk if untreated [13–15]. A systematic review showed that loneliness increases the risk of depression and stress [16]. Loneliness during COVID-19 has been found to worsen mental health by increasing anxiety, depression, interpersonal problems, and substance use [17, 18]. Students with loneliness concerns already experience increased academic challenges and are more susceptible to disengagement and attrition from their studies [19–21]. Loneliness also causes time management problems, leads to uncontrolled internet addiction, and influences academic performance among students [22, 23]. Students with preexisting mental health concerns and loneliness may be at greater risk for heightened psychological distress stemming from COVID-19 compared to students without such concerns and loneliness [24, 25]. For these individuals, increased loneliness could exacerbate existing symptoms and lead to episodic relapses of mental illness [24–27].

Job security, participation in physical and social activities, connection with others in the context of one's job, and loss of the care and support provided by professionals have been found to increase the sense of perceived risk associated with the pandemic and exacerbate loneliness [28]. A recent study suggested that young adults may be disproportionately affected by disease containment policies that increase social isolation and the risk of loneliness [29]. Another recent studies also suggested that younger adults may be at increased risk for distress and loneliness during COVID-19, relative to older adults [30, 31]. Several additional factors were also identified as being associated with loneliness in previous studies conducted in different parts of the globe. These include factors such as female sex [7, 11, 32–37], male sex [38], age [39, 40], economic status [10, 41–44], non-health department [8], sleeping problem [10, 45, 46], perceived stress [47], poor social support [10, 43, 48–51], and substance abuse [49, 52, 53].

Higher education institutions took measures including quarantines, physical distancing, and closing universities to reduce the transmission of the virus and slow the spread of the pandemic [54–57]. The challenges university students were experiencing in response to COVID-19 could result in campus closures, disruptions to research and internship placements, and exam cancellations [18, 56, 58, 59].

The online and digital platforms are typically used in high-income countries to uphold social connections and avoid prohibitions on in-person contact, which prevent loneliness [60]. However, the use of digital platforms in developing countries, including Ethiopia, is minimal due to low internet infrastructure. Worries about contracting the disease and fears of infecting loved ones can lead to staying at home with family members. This reduction in the frequency of social contact represents an extreme disruption to social life and can enhance loneliness [61]. As many universities suspended classroom teaching and switched to online teaching, this resulted in social distancing measures [62], which may affect their psychological well-being and mental health, including loneliness [63]. The decline in personal social contact due to COVID-19 has resulted in a heightened level of loneliness [64].

There is a lack of evidence about loneliness among university students in general [65] and during COVID-19 time in particular, in Ethiopia. Collecting data regarding loneliness across university students could be of great importance in determining the prevalence and the determinant factors that contribute to loneliness. The study will help the stakeholders to design appropriate interventions based on the results of the study. Therefore, the current study aims to determine loneliness and associated factors among university students in Ethiopia. The investigation will signal the stakeholder groups to address possible mechanisms to combat the problems of loneliness imposed by COVID-19.

## Methods

### Study design, setting and period

A cross-sectional study design was used among university students in Ethiopia. This study design was chosen because it is relatively fast and inexpensive for population-based surveys to assess the prevalence of loneliness. The data were collected from May 30 to June 30, 2021.

### Study population and eligibility criteria

All university students who had used social media (such as Telegram, Facebook, and Imo) and who were above the age of 18 years were included. University students who were willing to participate and were available online during the study period were enrolled. We used the snowball sampling technique to access university students who were using social media. The survey was voluntary based, and a participant consent form was attached to the online instruments at the beginning of the questionnaire. Students were asked to continue the survey once they read the introduction of the questionnaire, which included the purpose of the study, consent to participate, and the confidentiality issue, as well as the ability to discontinue even if they started to fill it out. The flow chart of the study participants is as follows, including the year of study (Fig 1).

### Sampling technique and sample size determination

The sample size was 426, calculated based on the previous study in Ethiopia (49.5% prevalence of loneliness) using the single population proportion formula by adding a non-response rate of 10% [65]. The snowball sampling technique was used in the current study. We have used this technique to easily access students to share the data collection tools.

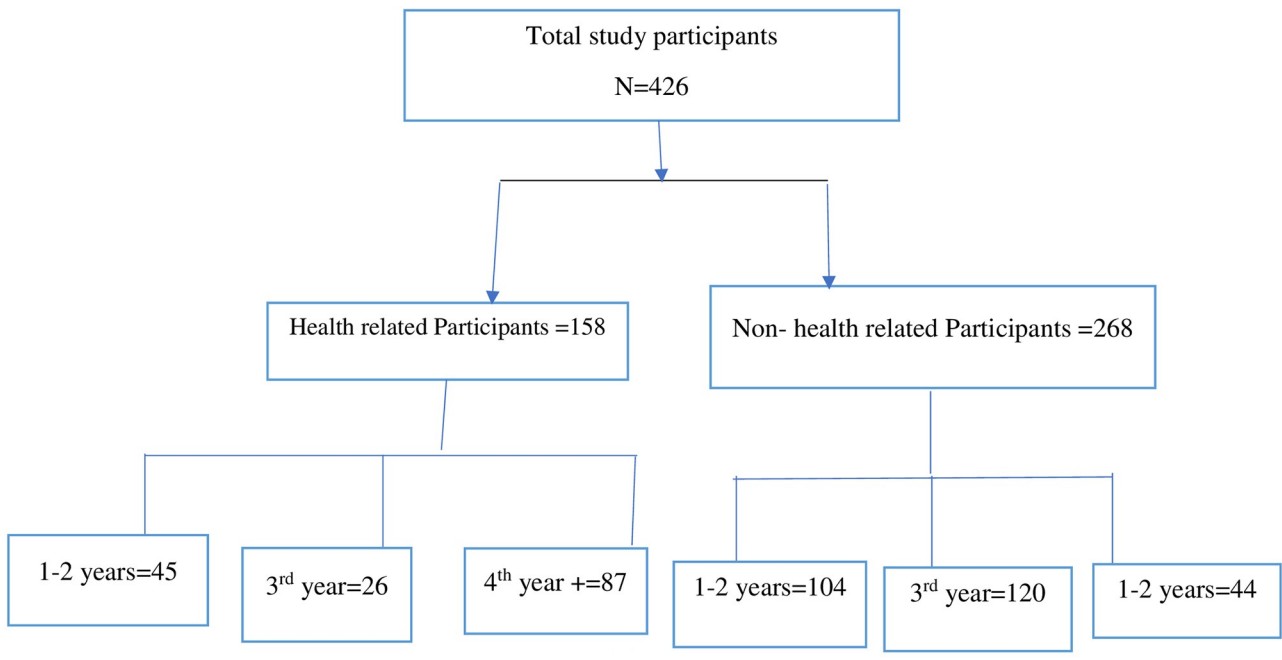

**Fig 1. Schematic presentation of study participants.**

## Data collection tool and procedure

The data collection tool used for the present study consists of three parts. The first part consists of questions on socio-demographic characteristics. The second part consists of questions to assess social support and loneliness. Social support was assessed by the Oslo 3-item Social Support Scale (OSSS-3). Loneliness was measured by the University of California Los Angeles Loneliness Scales (UCLA-8). The third part consists of a perceived stress scale (PSS-10) to assess perceived stress among university students. The questionnaire was distributed to social media users among university students via a telegram group, emails sent through a student representative, and Facebook.

## Measurement of variables

**Loneliness.**   The UCLA-8 was used to measure loneliness [42, 66]. Scores from the eight-item scale were categorized by degrees of loneliness: none (8–16), mild (17–20), moderate (21–24), or severe (> 24) [67, 68]. Finally, those who reported mild, moderate, and severe loneliness were categorized as having loneliness.

**Sexual harassment.**   Respondents were asked if they had ever encountered any form of sexual harassment. Those who encountered at least one form of sexual harassment were considered to have experienced sexual harassment.

**Smoking and alcohol use status.**   Respondents were asked if they have ever smoked cigarettes and alcohol in their life time.

**Perceived stress.**   PSS-10 was used to measure perceived stress level of students. The tool is validated in Ethiopia [69]. A cut of point of $\geq 20$ was considered as encountering stress.

**Self-efficacy.**   Students were asked students using a single item to assess the levels of their self-efficacy related to COVID-19 by "How confident are you that you can prevent getting COVID-19 in case of an outbreak?" [70].

**Social support.** Oslo 3-items Social Support Scale (OSSS-3) was used to assess the level of social support received [71]. Score 3 to 8 were considered as poor, scores 9 to 11 as moderate and 12 to 14 indicates strong social support [72].

**Sleeping problems.** Sleeping problems (or trouble sleeping) were defined as present or not during the last two weeks before the interview [73].

## Statistical analysis

The data were collected through online Google Forms and exported to SPSS version 26 for analysis. The means, frequencies, and percentages were computed. Logistic regression was used to identify factors associated with loneliness. Independent variables having a p-value less than 0.2 with a dependent variable (loneliness) were candidates for multivariable regression. Those independent variables having less than 0.05 p-values were judged to be factors for loneliness.

## Ethics approval and consent to participate

Letter of permission was taken from University of Gondar School of pharmacy Institutional Review Board with ethical clearance reference number of SOP 143/20. Written consent to participate was sent to each participant with online form and those study participants who were voluntary to participate had respond to the questionnaire. Before respondents were requested for consent, they were informed well about the purpose and potential benefits of the study; the confidentiality of the information collected from them, and their full right not to give a response to specific questions, or not to participate at all. Written consent was received from each participant using electronic signature to assure their willingness of participation and no identifiers were listed in the questionnaire to make it confidential. This study was conducted according to the declaration of Helsinki.

## Results

### Socio-demographic characteristics

In this study, four hundred twenty-six university students participated, of which two hundred and sixty-eight (62.9%) were males. The mean age of the respondents was 23.5 years (±3.42). About 37.1% of participants were from health-related departments (Table 1).

### Loneliness and related information

Two hundred and sixty-three (61.7%) participants had self-reported sleeping problems during COVID-19, and about 31.7% reported being extremely susceptible to the disease. About seventy-eight study participants (18.3%) had perceived stress, and two hundred and seventy-seven students (65.5%) had poor social support. In the current study, three hundred and twenty-six (76.5%, 95% CL: 72.3–80.5) university students had symptoms of loneliness (Table 2).

### Factors associated with loneliness during COVID-19

In the current study, sex, residence, department, living alone, sexual harassment, sleeping problem, having chronic disease, feeling extremely susceptible to COVID-19, self-efficacy, perceived stress, and social support were candidate variables for the final model (p-value < 0.2), and entered into multivariable logistic regression. In the final model, being female (AOR = 1.75; 95% CI: 1.01, 3.04), non-health-related departments (AOR = 1.94; 95% CI: 1.17, 3.35), ever encountered sexual harassment (AOR = 3.32; 95% CI: 1.46, 7.53), self-reported

**Table 1. Socio-demographic characteristics of study participants among university students in Ethiopia, 2021 (n = 426).**

| Variable | Categories | Frequency | Percent |
|---|---|---|---|
| Sex | Female | 158 | 37.1 |
| | Male | 268 | 62.9 |
| Age | 18–22 | 179 | 42.0 |
| | 23–37 | 247 | 58.0 |
| Department | Health related | 158 | 37.1 |
| | Not health related | 268 | 62.9 |
| Residence | Rural | 153 | 35.9 |
| | Urban | 273 | 64.1 |
| Years of study | 1–2 years | 149 | 35.0 |
| | 3$^{rd}$ year | 146 | 34.3 |
| | 4$^{th}$ + years | 131 | 30.8 |
| Love engagement | No | 301 | 70.7 |
| | Yes | 125 | 29.3 |
| Living alone | Yes | 134 | 31.5 |
| | No | 292 | 68.5 |

**Table 2. Loneliness and related information of study participants among university students in Ethiopia, 2021 (n = 426).**

| Variable | Categories | Frequency | Percent |
|---|---|---|---|
| Ever encountered sexual harassment | No | 316 | 74.2 |
| | Yes | 110 | 25.8 |
| Smoking | No | 383 | 89.9 |
| | Yes | 43 | 10.1 |
| Chat chewing | No | 355 | 83.3 |
| | Yes | 71 | 16.7 |
| Alcohol drink | No | 231 | 54.2 |
| | Yes | 195 | 45.8 |
| Sleeping problem | No | 263 | 61.7 |
| | Yes | 163 | 38.3 |
| Chronic disease | No | 374 | 87.8 |
| | Yes | 52 | 12.2 |
| Extreme susceptibility to COVID-19 | No | 291 | 68.3 |
| | Yes | 135 | 31.7 |
| Do you daily talk about COVID-19 | No | 285 | 66.9 |
| | Yes | 141 | 33.1 |
| Self-efficacy | Not self-efficacious | 287 | 67.4 |
| | Yes self-efficacious | 139 | 32.6 |
| Do you check COVID -19 is report daily | No | 304 | 71.4 |
| | Yes | 122 | 28.6 |
| Perceived stress | No | 348 | 81.7 |
| | Yes | 78 | 18.3 |
| Social support | Poor | 277 | 65.0 |
| | Moderate | 124 | 29.1 |
| | Strong | 25 | 5.9 |
| Loneliness | Yes | 326 | 76.5 |
| | No | 100 | 23.5 |

**Table 3. Associated factors of loneliness, among university students in Ethiopia, 2021 (n = 426).**

| Variables | Categories | Loneliness | | COR (95% UI) | AOR (95% UI) |
|---|---|---|---|---|---|
| | | Yes (%) | No (%) | | |
| Sex | Male | 195(72.8) | 73(27.2) | 1 | 1 |
| | Female | 131(82.9) | 27(17.1) | 1.82(1.11,2.98) | 1.75(1.01,3.04) * |
| Residence | Urban | 198(72.5) | 75(27.5) | 1 | 1 |
| | Rural | 128(83.7) | 25(16.3) | 1.94(1.17,3.21) | 1.43(0.81,2.54) |
| Department | Health-related | 107(67.7) | 51(32.3) | 1 | 1 |
| | Not health-related | 219(81.7) | 49(18.3) | 2.13(1.35,3.35) | 1.94(1.17,3.35) * |
| Living alone | No | 215(73.6) | 77(26.4) | 1 | 1 |
| | Yes | 111(82.8) | 23(17.2) | 1.73(1.03,2.90) | 1.61(0.88,2.94) |
| Ever encountered sexual harassment | No | 225(71.2) | 91(28.8) | 1 | 1 |
| | Yes | 101(91.8) | 9(8.2) | 4.54(2.20,9.36) | 3.32(1.46,7.53) ** |
| Sleeping problem | No | 177(67.3) | 86(32.7) | 1 | 1 |
| | Yes | 149(91.4) | 14(8.6) | 5.17(2.82,9.47) | 2.13(1.06,4.30) * |
| Chronic disease | No | 278(74.3) | 96(25.7) | 1 | 1 |
| | Yes | 48(92.3) | 4(7.7) | 4.14(1.46,11.80) | 2.31(0.74,7.23) |
| Extremely susceptible to COVID-19 | Yes | 112(83.0) | 23(17.0) | 1.75(1.04,2.94) | 1.06(0.58,1.95) |
| | No | 214(73.5) | 77(26.5) | 1 | 1 |
| Self-efficacy | No | 231(80.5) | 56(19.5) | 1.91(1.20,3.03) | 1.35(0.79,2.30) |
| | Yes | 95(68.3) | 44(31.7) | 1 | 1 |
| Perceived stress | Yes | 75(96.2) | 3(3.8) | 9.66(2.98,31.36) | 6.40(1.85,22.19) ** |
| | No | 251(72.1) | 97(27.9) | 1 | 1 |
| Social support | Strong | 13(52.0) | 12(48.0) | 1 | 1 |
| | Moderate | 95(76.6) | 29(23.4) | 3.55(1.46,8.62) | 2.47(0.88,6.90) |
| | Poor | 219(79.1) | 58(20.9) | 4.09(1.77,9.44) | 3.13(1.10,8.87) * |

Hosmer and Lemeshow goodness of fit p = 0.741,

* p<0.05 and

** p<0.01

sleeping problem (AOR = 2.13; 95% CI: 1.06, 4.30), perceived stress (AOR = 6.40; 95% CI: 1.85, 22.19), and poor social support (AOR = 3.13; 95% CI: 1.10, 8.87) were significantly associated with loneliness (Table 3).

## Discussion

The present study assessed loneliness and associated factors among university students in Ethiopia during the COVID-19 pandemic through an online survey (using social media platforms such as Telegram, Facebook, and Imo). The prevalence of loneliness in the current study was 76.5%, with a 95% CI: 72.3%, 80.5%. Being female, non-health-related departments, ever encountered sexual harassment, having a self-reported sleeping problem, perceived stress, and poor social support were significantly associated with loneliness.

The prevalence of loneliness in the present study is higher than in other previous studies conducted elsewhere [7, 8, 10, 34, 43, 49, 52, 74]. The variation might be due to differences in methods and the tools used. For example, in the current study, we have used the UCLA-8 to measure loneliness, whereas a recent study in the United Kingdom [10] and a study in Norway have used the Three-Item Loneliness Scale (TILS). Other factors, such as socio-cultural difference and study period, might contribute to the variation. The current prevalence was much

higher than in a report prior to the pandemic, as evidenced by the previous study [65]. This is not surprising, as the impact of COVID-19 would result in a higher prevalence of loneliness [75]. A higher level of loneliness may result in mental health problems, as evidenced by a rapid systematic review [5] and studies showed loneliness to be related to negative mental health symptoms [76, 77].

In this study, the odds of loneliness among females were about 1.75 times higher than those among males. This is supported by earlier studies [7, 11, 32–37]. The reason might be due to the fact that females are more sensitive to emotional expressions, which is relevant to the antecedents of loneliness [78, 79]. In contrast to the current study, research conducted in Turkey confirmed that males frequently suffered from loneliness [38]. Other research revealed the absence of sex differences for the risk of loneliness [10, 40, 80, 81]. This might be due to other extraneous factors contributing to the variation.

Study participants from non-health-related departments had 1.94 times higher odds of developing loneliness than study participants from health-related departments. This aligns with another study [8]. The reason might be that non-health-related students developed a higher grade of psychological stress than health-related students during an outbreak of COVID-19 [82, 83].

In the current study, university students who ever encountered sexual harassment had three times greater risk for loneliness. Other evidence also indicate that harassed female students mostly experience symptom of loneliness and other mental health problems [84]. This might be due to sexual harassment has the effect of reducing one's own commitment to relationship [85].

Participants who reported sleeping problem was significantly associated with loneliness in the current study. Study participants who had sleeping problem had 2.14 times higher risk of loneliness than their counterparts. This aligns with studies done in Greece, France and UK population during COVID-19 pandemic [10, 45, 46]. Whether loneliness resulted the sleeping problem or the sleeping problem resulted in loneliness is not known in the current study as one of the inherent limitation of cross-sectional study is the "egg-chicken dilemma" [9, 86].

The current study identified perceived stress as a significant factor for loneliness. Study participants with symptoms of perceived stress had 6.4 time higher risk of loneliness than the counterparts. This is supported by a previous study [47]. This might be due to negative impact of lifetime stress exposure on mental health [87].

Poor social support was significantly associated with loneliness. Study participants with poor social support had 3.14 times higher risk of loneliness than those study participants reported to have good social support. This is consistent with previous studies [10, 43, 48–51]. This might be due limited support from interpersonal relationships [16].

The current study assessed substance use such as alcohol use, chat chewing, and cigarette smoking. In the regression model, such variables were not significantly associated with loneliness. However, alcohol use was significantly associated with loneliness in the other studies [49, 52]. In contrast to this study, other study finding revealed that students with loneliness have been found to have low alcohol consumption as compared to others without loneliness [88]. Cigarette smoking also explained having relation with loneliness of which one can affect the other and vice-versa [53].

In the current study living alone was not significantly associated with loneliness even though it was considered a candidate variable for multiple logistic regression. However, it was significantly associated with loneliness in the other previous study [7, 42]. The chronic disease status was not significantly associated with loneliness in the current study. However, it was significantly associated with several other studies [89].

This study has several limitations like social desirability bias as it was cross sectional study. As we have used online snowball sampling technique, university students who did not have smart phone or computer with internet access were not included which might affect the generalizability of the finding. Beyond the presence of COVID -19, loneliness might be caused by different reasons such as heritability and gene-type as explained in earlier genetics studies [51, 90, 91]. However, the current study had not assessed the genetic and hereditary factors that might influence loneliness among University students. Within such limitations, our finding provide baseline data on the prevalence of loneliness and identified associated factors signaling the stakeholders to take action to the problems.

## Conclusion

This study showed the higher prevalence of loneliness in the university students during COVID-19. The identified factors were being female, non-health department, ever encountered sexual harassment, and self-reported sleeping problem, perceived stress, and poor social support. These findings would suggest the stake holders to provide student counselling service, social support and psychological interventions to reduce loneliness and prevent related mental health problems among University students.

## Supporting information

**S1 Dataset.**
(SAV)

## Acknowledgments

The authors are grateful for the University of Gondar, and study participants.

## Author Contributions

**Conceptualization:** Wudneh Simegn, Gashaw Sisay, Henok Dagne.

**Data curation:** Wudneh Simegn, Gashaw Sisay, Abdulwase Mohammed Seid.

**Formal analysis:** Wudneh Simegn, Abdulwase Mohammed Seid, Henok Dagne.

**Funding acquisition:** Wudneh Simegn, Henok Dagne.

**Investigation:** Wudneh Simegn, Gashaw Sisay, Henok Dagne.

**Methodology:** Wudneh Simegn, Henok Dagne.

**Project administration:** Wudneh Simegn.

**Resources:** Wudneh Simegn, Henok Dagne.

**Software:** Wudneh Simegn, Henok Dagne.

**Supervision:** Henok Dagne.

**Validation:** Wudneh Simegn, Gashaw Sisay, Abdulwase Mohammed Seid, Henok Dagne.

**Visualization:** Henok Dagne.

**Writing – original draft:** Wudneh Simegn, Abdulwase Mohammed Seid, Henok Dagne.

**Writing – review & editing:** Wudneh Simegn, Gashaw Sisay, Abdulwase Mohammed Seid, Henok Dagne.

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
