## [Decision Letter · Decision Letter 0]

9 Mar 2023

PONE-D-22-01565Loneliness and its associated factors among University students during late stage of COVID-19 Pandemic: An online cross-sectional studyPLOS ONE

Dear Dr. Simegn Belay,

Thank you for submitting your manuscript to PLOS ONE. After careful consideration, we feel that it has merit but does not fully meet PLOS ONE’s publication criteria as it currently stands. Therefore, we invite you to submit a revised version of the manuscript that addresses the points raised during the review process.

We look forward to receiving your revised manuscript.

Kind regards,

Daniel Ahorsu, PhD

Academic Editor

PLOS ONE

Journal Requirements:

4. Please include a caption for figure 1.

Reviewers' comments:

Reviewer's Responses to Questions

**Comments to the Author**

1. Is the manuscript technically sound, and do the data support the conclusions?

Reviewer #1: Yes

Reviewer #2: Yes

2. Has the statistical analysis been performed appropriately and rigorously? 

Reviewer #1: Yes

Reviewer #2: I Don't Know

3. Have the authors made all data underlying the findings in their manuscript fully available?

Reviewer #1: Yes

Reviewer #2: No

4. Is the manuscript presented in an intelligible fashion and written in standard English?

Reviewer #1: Yes

Reviewer #2: Yes

5. Review Comments to the Author

Reviewer #1: Loneliness and its associated factors among University students during late stage of COVID-19 Pandemic: An online cross-sectional study titled has fulfilled all the basic and to extent survey made out of loneliness in the pandemic situation. There are lot of interlocks and chain of reference which found to be disgusting during the reading of the manuscript, that could be avoided. The SPSS survey results could be added as a pictorial representation to figure out the exact result in a better understanding way.

Reviewer #2: The authors did not submit their data with the written article submitted. The data analysis done was not presented in any format such as tabular format. However, they reported briefly of the findings of the written article, hence the reviewer is not able to review whether the information provided as the data analysis was adequately performed or not. It would be relevant for the authors of this written article to present a tabular format of the results of the data analysis, or if they can present the result of the analysis as an appendix

6. PLOS authors have the option to publish the peer review history of their article (what does this mean?). If published, this will include your full peer review and any attached files.

Reviewer #1: No

Reviewer #2: No

---

## [Author Response · Author response to Decision Letter 0]

13 Apr 2023

Response to Reviewers comments 

Point by point response for editors/reviewers’ comments Manuscript title: Loneliness and its associated factors among University students during late stage of COVID-19 Pandemic: An online cross-sectional study Manuscript ID: PONE-D-22-01565Dear editor/reviewers. Thank you for giving us the chance to revise the manuscript again. Saying this we addressed all the concerns raised by the reviewers and incorporated the authors’ reflection in the revised manuscript. Summary and General ImpressionThe authors will need to provide more in-depth analysis of the data to elaborate on the discussion a bit further Authors’ response: Thank you, the authors performed descriptive statistics and logistic regression. We included the result and we improved the discussion in the revised manuscript. The authors will need to explain the relevance of the studyAuthors’ response: we improved the relevant of the study in the revised manuscript. Further explanations should be provided under the discussion.Authors’ response: we improved the discussion thoroughly in the revised manuscript. Specific Comment from ReviewerAbstract: Conclusion. & Recommendation. L2- rephrase statement,Authors’ response: we revised the statement as “Being female, working in non-health-related fields, having sleeping problems, encountering sexual harassment, perceived stress, and poor social support were significantly associated with loneliness.”Introduction- Authors should rephrase statementsP.2: The higher can be removed…. Higher institutions… P.2, L3: rephrase statements. It is long.P. 3, L10 -sentence is long and windingP. 3-L8- may successfully have reduced… but which…P.4- could paragraph 4 move up right after paragraph 1? Otherwise, the introduction looks disjointed, following this paragraph.Authors’ response: Thank you. We rewrite all above recommendations as suggested (See the revised manuscript). MethodologyDesign- Authors should explain briefly the relevance of the cross-sectional design to the study. Tenses in the sentence should be relooked at, … design was… instead of …has been…Authors’ response: thank you. We amended as “A cross-sectional study design was used among university students in Ethiopia. This study design was chosen because it is relatively fast and inexpensive for population-based surveys to assess the prevalence of loneliness” in the revised manuscript. Study population- L.2- Rephrase sentence, Participation by university students was voluntary, they granted ethical consent….Authors’ response: Thank you. We revised as “University students who were willing to participate and were available online during the study period were enrolled” in the revised manuscript.Eligibility criteria should include age of participants. Age criteria particularly the minimum age was not stated. This is important to ensure that participants gave appropriate consent to participate in the study. Participants below 18years would need parental consent as well as assent. Authors’ response: Thank you. As per the authors know there were no university students below 18 years in Ethiopia. Students start their primary education at the age of 7 and for 12 years up to joining the university. This indicates 18 year is the minimum age for university students. However we accept the reviewers' concern that there may be possibility of this and to make it clear, we included the age as inclusion criteria as suggested “All university students who had used social media (such as Telegram, Facebook, and Imo) and who were above the age of 18 years were included.”Sampling technique- Rephrase second sentence L2).Authors’ response: Thank you, we rewrite as “The snowball sampling technique was used in the current study.”Data collection and procedure- Authors should mention the full names of tools administered and subsequently use the initials. It would be important that authors provide names of tools used,Authors’ response: Thank you. We included the full names of the tools in the revised manuscript. ResultSocio demographic Characteristics were provided but not details in a tabular for although authors stated table 1. Authors’ response: Thank you. Tables were separately submitted. We included tables in the main document (table 1 for sociodemographic characteristics, table 2 loneliness and related information and table 3 for associated factors) in the revised manuscript. DiscussionRephrase statements and some phrases, e.g., P.8 participants who reported sleeping problems instead of self-reported sleeping problemAuthors’ response: thank you. We revised the sentences as suggested “Participants who reported sleeping problem was significantly associated with loneliness in the current study”. Please some discussion may need further elaboration. E.g., the odds of loneliness among females being higher would need further explanation. Authors stated … females’ emotion being more sensitive and fragile… as reason however, the study to support that did not find any difference between the male and female gender. Authors’ response: we appreciate the reviewers concern on this statement that was references error. We remove this citation and rewrite and cite the statement gain in the revised manuscript. “This might be due to the fact that females are more sensitive to emotional expressions which is relevant to the antecedents of loneliness (77, 78).”. (https://doi.org/10.3389/fnhum.2018.00275 and doi:10.1093/geront/gnaa082 )P. 9- Did the study result show students from non-health department developing higher levels of psychological stress?Authors’ response: thank you. The result did not assessed the stress among the two groups. But the authors standing point to this explanation was the previous studies that have been cited. We also confirmed that perceived stress was associated with loneliness in the current study as indicate in table 2. P.10- harassed female students mostly experience feeling of anger not feeling and anger. 2. Did this study measure feeling of anger? How is this substantiated? Authors’ response: Thank you, the current study did not assessed anger. We improved this statement as “Harassed female students mostly experience symptom of loneliness and other mental health problems”. P. 12= this cannot be a paragraph because it is just one statement. Further explanation would help substantiate this finding.Authors’ response: Thank you the authors improved this paragraph ay adding the odds ratio and justification. “The current study identified perceived stress as a significant factor for loneliness. Study participants with symptoms of perceived stress had 6.4 time higher risk of loneliness than the counterparts. This is supported by a previous study (47). This might be due to negative impact of lifetime stress exposure on mental health (86).”. Reviewer #1: Loneliness and its associated factors among University students during late stage of COVID-19 Pandemic: An online cross-sectional study titled has fulfilled all the basic and to extent survey made out of loneliness in the pandemic situation.There are lot of interlocks and chain of reference which found to be disgusting during the reading of the manuscript that could be avoided. Authors’ response: Thank you, the authors seen the whole document and amend some errors (see the track changes). The SPSS survey results could be added as a pictorial representation to figure out the exact result in a better understanding way.Authors’ response: Thank you, the authors added the SPSS survey results (Table 1 and 2) in the revised manuscript. Reviewer #2: The authors did not submit their data with the written article submitted. Authors’ response: Thank you. The authors added the data as supplementary file. The data analysis done was not presented in any format such as tabular format. However, they reported briefly of the findings of the written article, hence the reviewer is not able to review whether the information provided as the data analysis was adequately performed or not. Authors’ response: Thank you, the authors added the SPSS survey results (Table 1 and 2) in the revised manuscript.It would be relevant for the authors of this written article to present a tabular format of the results of the data analysis, or if they can present the result of the analysis as an appendix.Authors’ response: Thank you, the authors added the SPSS survey results with in the main document (Table 1 and 2) in the revised manuscript and as supplement file.

---

## [Decision Letter · Decision Letter 1]

5 Jun 2023

Loneliness and its associated factors among University students during late stage of COVID-19 Pandemic: An online cross-sectional study

PONE-D-22-01565R1

Dear Dr. Simegn Belay,

We’re pleased to inform you that your manuscript has been judged scientifically suitable for publication and will be formally accepted for publication once it meets all outstanding technical requirements.

Kind regards,

Daniel Ahorsu, PhD

Academic Editor

PLOS ONE

Additional Editor Comments (optional):

Reviewers' comments:

Reviewer's Responses to Questions

**Comments to the Author**

1. If the authors have adequately addressed your comments raised in a previous round of review and you feel that this manuscript is now acceptable for publication, you may indicate that here to bypass the “Comments to the Author” section, enter your conflict of interest statement in the “Confidential to Editor” section, and submit your "Accept" recommendation.

Reviewer #1: All comments have been addressed

Reviewer #2: All comments have been addressed

2. Is the manuscript technically sound, and do the data support the conclusions?

Reviewer #1: Yes

Reviewer #2: Yes

3. Has the statistical analysis been performed appropriately and rigorously? 

Reviewer #1: Yes

Reviewer #2: Yes

4. Have the authors made all data underlying the findings in their manuscript fully available?

Reviewer #1: Yes

Reviewer #2: Yes

5. Is the manuscript presented in an intelligible fashion and written in standard English?

Reviewer #1: Yes

Reviewer #2: Yes

6. Review Comments to the Author

Reviewer #1: work has been justified and relevant comments have been addressed though it can be considered further

Reviewer #2: the Authors have made all the appropriate corrections in their revision. And the manuscript have been written in a coherent, and in standard English language, which is understood. In addition, data was provided for the reviewer and other results were presented in the work.

Generally, the authors have made the necessary input for this manuscript to be accepted.

7. PLOS authors have the option to publish the peer review history of their article (what does this mean?). If published, this will include your full peer review and any attached files.

Reviewer #1: No

Reviewer #2: No

---

## [Editor Report · Acceptance letter]

26 Jun 2023

PONE-D-22-01565R1 

Loneliness and its associated factors among University students during late stage of COVID-19 Pandemic: An online cross-sectional study 

Dear Dr. Simegn:

I'm pleased to inform you that your manuscript has been deemed suitable for publication in PLOS ONE. Congratulations! Your manuscript is now with our production department. 

Kind regards, 

on behalf of

Dr. Daniel Ahorsu 

Academic Editor

PLOS ONE